# SCoRe: Pre-Training for Context Representation in Conversational Semantic Parsing

**Tao Yu**
Yale University
`tao.yu@yale.edu`

**Rui Zhang**
The Pennsylvania State University
`rmz5227@psu.edu`

**Oleksandr Polozov, Christopher Meek, Ahmed Hassan Awadallah**
Microsoft Research
`{polozov,meek,hassanam}@microsoft.com`

## Abstract

Conversational Semantic Parsing (CSP) is the task of converting a sequence of natural language queries to formal language (e.g., SQL, SPARQL) that can be executed against a structured ontology (e.g. databases, knowledge bases). To accomplish this task, a CSP system needs to model the relation between the unstructured language utterance and the structured ontology while representing the multi-turn dynamics of the dialog. Pre-trained language models (LMs) are the state-of-the-art for various natural language processing tasks. However, existing pre-trained LMs that use language modeling training objectives over free-form text have limited ability to represent natural language references to contextual structural data. In this work, we present SCoRe, a new pre-training approach for CSP tasks designed to induce representations that capture the alignment between the dialogue flow and the structural context. We demonstrate the broad applicability of SCoRe to CSP tasks by combining SCoRe with strong base systems on four different tasks (SParC, CoSQL, MWoZ, and SQA). We show that SCoRe can improve the performance over all these base systems by a significant margin and achieves state-of-the-art results on three of them.

## 1 Introduction

The goal of task-oriented dialog systems is to assist the user in completing a certain task by performing an action or retrieving relevant information (Tur & Mori, 2011). They are often built on top of a structured ontology grounded in a knowledge base, a database, or a set of API calls. This in contrast to open-domain dialog systems (also referred to as chit-chat systems) where the goal is to maximize engagement with users in open-ended conversations (Jafarpour et al., 2010; Ritter et al., 2011).

A key component of task-oriented conversational systems is Conversational Semantic Parsing (CSP), which converts each utterance in the dialog into a formal language query (e.g., SQL, SPARQL) that can be executed against the structured ontology. CSP has been extensively studied in several academic and industrial research settings such as dialog systems (e.g., dialog state tracking in MWoZ (Budzianowski et al., 2018)), interacting with physical agents (e.g., (Chai et al., 2018)), context-dependent semantic parsing (e.g., SParC (Yu et al., 2019b)), SQL-grounded state tracking (e.g., CoSQL (Yu et al., 2019a)), and sequential question answering (e.g., SQA (Iyyer et al., 2017)). These settings differ in some respect, but they share the same overall objective and key challenge: *how to jointly represent the natural language utterances and underlying structured ontology while taking into consideration the multi-turn dynamics of the dialog*.

Similar to many other natural language tasks, recent work in CSP has significantly benefited from advances in language model pre-training. However, existing general-purpose pre-trained language models, e.g. BERT (Devlin et al., 2019), are pre-trained on free-form text data using language model objectives. This limits their ability in modeling the structural context or the multi-turn dynamics of the dialogs. This presents an opportunity to improve pre-trained LMs to specifically address these limitations for CSP tasks. Recent work has demonstrated the benefits of adapting pre-trained LMs

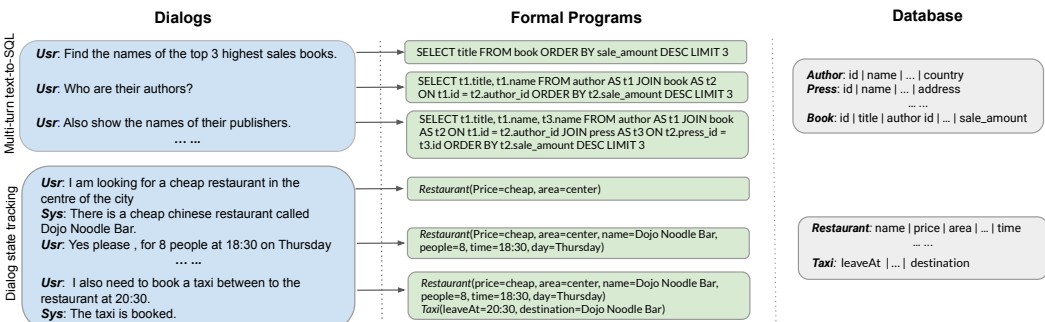

Figure 1: Examples of conversational semantic parsing tasks from SPARC and MWOZ datasets.

to specific domains (Gururangan et al., 2020) or tasks (Zhang et al., 2019b) via a second phase of pre-training. For example, open-domain dialogue language models such as DialoGPT (Zhang et al., 2020) and ConveRT (Henderson et al., 2019) are pre-trained on the Reddit data and applied to dialog response generation and retrieval tasks.

In this paper, we introduce SCORE (**S**tructured & **S**equential **Co**ntext **Re**presentation), a language model pre-training approach for CSP tasks. SCORE adapts general pre-trained LMs by introducing a second phase of pre-training using multiple objectives that capture both multi-turn dynamics and the structural contexts in a dialog. In contrast to open-domain dialogs, CSP datasets are usually much smaller due to the difficulty and expense of obtaining and labeling data (mapping natural language utterances to formal language). Unlike most prior work on contextualized LMs which are pre-trained on free text, according to the finding where questions in CSP tasks are more compositional than other free-text since they can be mapped into formal representations, we propose to train SCORE on synthesized conversational semantic parsing data with multiple training objectives that aim to ground utterances into the schema of the underlying ontology and to model the relationship between different utterances in the multi-turn conversation. In this way, SCORE can effectively inject structural and conversational inductive biases in LMs that can translate to many CSP tasks. SCORE uses an order of magnitude smaller dataset for the second stage of pre-training, does not require changes to the pre-trained model architecture, can be used as a drop-in replacement of general pre-trained LMs with any semantic parsing model, and can be used out-of-the-box in many CSP tasks.

We apply SCORE to four different CSP tasks: (1) sequential text-to-SQL (SPARC), (2) conversational text-to-SQL (COSQL), (3) dialog state tracking (MWOZ), and (4) weakly-supervised sequential question answering (SQA). The fours tasks represent different scenarios, types of ontologies, supervision signals, system responses, and domains (see Table 1 for a detailed comparison and Figure 1 for examples). We demonstrate that: (1) SCORE training objectives can effectively incorporate synthesized data, (2) a single pre-trained SCORE model can be used for several CSP tasks and can be combined with many baseline systems with different model architectures and (3) SCORE significantly improve all baseline systems and achieves new state-of-the-art results on three benchmarks (SPARC, SPARC, and MWOZ) and comparable performance to state-of-the-art results on the fourth (SQA).

## 2 APPROACH

The key challenge of CSP is to capture the relationship between the natural language utterance and the structured ontology in the multi-turn dialog dynamics. To this end, we inject structural and conversational inductive biases in SCORE by introducing two objective functions: *Column Contextual Semantics (CCS)* and the *Turn Contextual Switch (TCS)*. Because the size of existing semantic parsing datasets is limited, we produce synthesized data for pretraining SCORE by sampling from the context-free grammar induced from complex text-to-SQL examples in different domains. Moreover, to prevent SCORE from overfitting to the linguistic pattern of our synthesized data, we use the *Masked Language Modeling (MLM)* objective on human-generated utterances as regularization.

### 2.1 PRELIMINARIES

**Task Definition** In CSP, at each turn $t$, we aim to produce a formal representation $q_t$ given the current utterance $u_t$, the interaction history $h_t = [u_1, u_2, \ldots, u_{t-1}]$, and the schema $c$ (table and column names, slots, etc.) of the target database (ontology) $d$. To cover different problem variants, we

| Dataset | Structured Ontology | Annotation (Supervision) | Cross Domain | System Response | # Dialogs | # Turns |
|---------|---------------------|--------------------------|--------------|-----------------|-----------|---------|
| SPARC | database | SQL (supervised) | ✓ | ✗ | 4,298 | 12,726 |
| COSQL | database | SQL (supervised) | ✓ | ✓ | 3,007 | 15,598 |
| MWoZ | domain ontology | slot-value (supervised) | ✗ | ✓ | 8,438 | 113,556 |
| SQA | table | denotation (weakly-supervised) | ✓ | ✗ | 6,066 | 17,553 |

Table 1: Comparison of CSP datasets. Examples from two of the datasets are shown in Figure 1. Cross-domain means the train and test sets have different domains, so MWoZ is not cross-domain.

consider four popular CSP tasks shown in Table 1: SPARC (sequential text-to-SQL), COSQL (conversational text-to-SQL), MWoZ (dialogue state tracking), and SQA (weakly supervised sequential question answering). They have different target formal language and structured ontology:

- For the **utterance** $u$, it is the user question for SPARC and SQA, while for COSQL and MWoZ, $u$ is the combination of a user query and a system response.
- For the **database** $d$, SPARC and COSQL use multi-table databases; for MWoZ, the pre-defined ontology $d$ can also be viewed as a database; for SQA, $d$ is a single table.
- For the **formal representation** $q$, it is the SQL query for SPARC and COSQL; in MWoZ it is the slot-value pairs that can be viewed as simple SQL queries consisting of `SELECT` and `WHERE` clauses; and for SQA, $q$ is the latent program.

**Base Architecture**    The base architecture of SCORE takes as input a single turn of a CSP dialog $\langle u_t, h_t \rangle$ jointly with the underlying database schema $c$. Given this *contextualized conversational input* $C_t = \langle u_t, h_t, c \rangle$, SCORE encodes it into *contextualized conversation representations* $\vec{S}_t$ for each token in $C_t$. The encoder architecture follows RoBERTa (Liu et al., 2019b). It is then followed by a linear layer and normalized (Ba et al., 2016) to produce final representations $\vec{h}_t$ for each token:

$$C_t = \langle u_t, h_t, c \rangle, \ \vec{S}_t = \text{RoBERTa}(C_t), \ \boldsymbol{h}_{t,i} = \text{LayerNorm}(\text{GELU}(\boldsymbol{W_1} \boldsymbol{S}_{t,i})) \ \forall \ \boldsymbol{S}_{t,i} \in \vec{S}_t, \quad (1)$$

where GELU is an activation by Hendrycks & Gimpel (2016) and $\boldsymbol{W_1}$ is a learned parameter matrix.

To build $C_t$, we first concatenate current utterances $u_t$ and dialog history $h_t$ separated by a special token ``, as this simple strategy has been shown effective in state-of-the-art CSP systems (Zhang et al., 2019c; Wu et al., 2019; Liu et al., 2020; Heck et al., 2020). To incorporate the database schema, we follow Hwang et al. (2019) to concatenate all column names as a single sequence. Column names are separated by the special token `` and prefixed by their corresponding table name.

## 2.2 SCORE PRE-TRAINING

SCORE addresses the challenges of CSP by *pre-training a task-oriented language model contextualized by the conversational flow and the underlying ontology*. In pre-training, the SCORE model is self-supervised by two novel objectives in addition to the established Masked Language Modeling (MLM) objective. These objectives facilitate the accurate representation of the conversational flow between dialog turns and how this flow maps to the desired columns in the ontology.

**Column Contextual Semantics**    The first challenge of CSP is capturing the alignment between the natural language utterance and the underlying database schema. To address it, we optimize the SCORE model with the auxiliary objective of *Column Contextual Semantics (CCS)*. For each column in the database schema $c$, CCS targets the *operations* that should be performed on this column in a given conversational turn. Specifically, each formal representation $q$ is decomposed into operations on columns and tables, e.g. `GROUP BY` and `HAVING` for SQL queries, or `WHERE` for the slot-value pairs. In this way, our data covers 148 column operations. We use the encoding of the special token `` right before each column or table name to predict its corresponding operations, and then compute the CCS loss:

$$\mathcal{L}_{\text{CCS}}(C_t) = \sum\nolimits_{i \in c} \text{CrossEntropy}_{148}(\text{LayerNorm}(\boldsymbol{W_2} \, \boldsymbol{h}_{t,i}^c), \text{CCS}(q_t)) \quad (2)$$

where $\boldsymbol{h}_{t,i}^c$ is the contextualized representation of the $i^{\text{th}}$ column's special token `` in the contextualized input $C_t$, $\text{CCS}(q_t)$ returns the column operation label for the current formal representation $q_t$, $\text{CrossEntropy}_{148}$ computes the 148-way cross-entropy between the column operation prediction and label, and $\boldsymbol{W_2}$ is a learned parameter matrix.

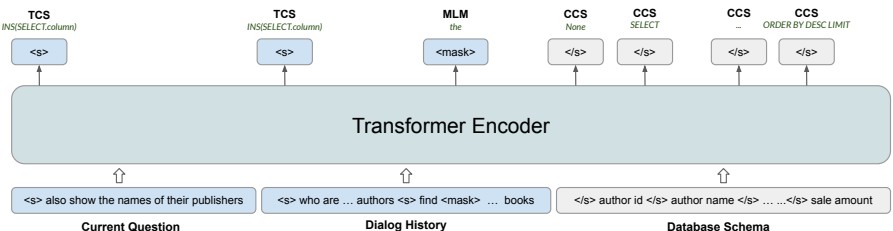

Figure 2: Pre-training of a SCORE encoder on a SPARC text-to-SQL example from Figure 1.

**Turn Contextual Switch**    The second challenge of CSP is capturing the conversational context flow and how it is grounded into the formal representations. The TCS objective aims to capture this grounding of context flow. To this end, it targets predicting *the difference in formal representations between dialog turns* based on the natural language utterance.

Based on the context-free grammar of SQL, we identify 26 possible *turn difference operations* that a conversational turn could elicit. They encode changes between different turns of user queries (the system response is not involved here) since we assume that most turn contextual shifts are from the user. For example, INS(WHERE) indicates inserting a new WHERE condition and DEL(SELECT.agg) indicates removing an aggregate operation from a SELECT statement (e.g. when an utterance *"Show all the ages instead."* elicits a change SELECT MAX(age) ... → SELECT age ...). We use the encoding of the special token  right before each turn to predict the context switch label between this turn and the previous history:

$$\mathcal{L}_{\text{TCS}}(C_t) = \text{CrossEntropy}_{26}(\text{LayerNorm}(\boldsymbol{W}_3\boldsymbol{H}_t^s), \text{TCS}(q_t, q_{t-1})) \tag{3}$$

where $\boldsymbol{H}_t^s \in \mathbb{R}^{(t-1)\times d}$ is the contextualized representation of all previous turns in $C_t$ with hidden dimension $d$, $\text{TCS}(q_t, q_{t-1})$ returns the turn difference operations from $q_{t-1}$ to $q_t$, and $\boldsymbol{W}_3$ is a learned parameter matrix. We don't use this objective to pre-train SCORE for MWOZ because the context switch label between turns is relatively simple in MWOZ (only select and where changes).

**Masked Language Modeling**    As in prior work on large-scale language models (Devlin et al., 2019), we use the *Masked Language Modeling (MLM)* objective to facilitate contextual representation learning for natural language utterances. Importantly for regularization, we only apply this loss on *in-domain human-annotated* natural language data. Namely, it includes utterances in SPARC, CoSQL, and SQA as well as nine task-oriented dialog datasets processed by Wu et al. (2020) for MWOZ (see data statistics in Figure 4). Formally, the MLM loss is given by:

$$\mathcal{L}_{\text{MLM}}(C_t) = \sum_m \text{CrossEntropy}_{\text{Vocab}}(\text{LayerNorm}(\boldsymbol{W}_4\boldsymbol{h}_t^m)) \tag{4}$$

where $\boldsymbol{h}_t^m$ are the contextualized representations of the masked 15% of tokens in $C_t$, and $\boldsymbol{W}_4$ is a learned parameter matrix.

**Pre-Training Setup and Steps**    To summarize the pre-training steps, we first collect a dataset $\mathcal{D}_{\text{nat}}$ of combined human-annotated natural language questions (without labels) from existing CSP tasks (as mentioned above), and create a large synthesized conversational data $\mathcal{D}_{\text{syn}}$ that is generated by a grammar induced from a small set of SPARC annotated examples (See 2.3). After that, we incorporate both two datasets in pre-training. More specifically, synthetic and natural examples are randomly sampled during pre-training. The total pre-training loss is the sum of the three objectives with CCS and TCS only applied to $\mathcal{D}_{\text{syn}}$ and MLM only to $\mathcal{D}_{\text{nat}}$:

$$\mathcal{L} = \sum_{C_t \in \mathcal{D}_{\text{syn}}} (\mathcal{L}_{\text{CCS}}(C_t) + \mathcal{L}_{\text{TCS}}(C_t)) + \sum_{C_t \in \mathcal{D}_{\text{nat}}} \mathcal{L}_{\text{MLM}}(C_t) \tag{5}$$

Figure 2 shows an overview of SCORE pre-training on an example SPARC dialogue from Figure 1. We report additional implementation details for pre-training SCORE in Section 3.3 and Appendix C.

## 2.3 Data Synthesis

We re-use the synthetic dataset of 120k synthetic task-oriented dialogues for MWOZ, introduced by Campagna et al. (2020). In this work, we introduce a complementary procedure to synthesize data for conversational text-to-SQL dialogues. We use about 400k tables in WIKITABLES (Bhagavatula et al., 2015) (after filtering and cleaning), WikiSQL, and Spider datasets as underlying databases $d$, and then synthesize about one dialog for each table. Finally, we synthesize 435k text-to-SQL conversations in total. Table 12 in Appendix B shows an example of the synthesized question-SQL pairs and their corresponding templates in our grammar.

To this end, we use only 500 dev examples from SPARC to induce two utterance-SQL generation grammars: (1) a single-turn context-free grammar $G_s$ for generating context-independent question-SQL pairs, and (2) a follow-up context-free grammar $G_c$ for follow-up question-SQL pairs. The single-turn grammar $G_s$ contains a list of synchronous question-SQL templates where typed slots (COLUMN0, OP0, VALUE0, . . . ) represent mentions of tables, columns, values, and SQL operations. The follow-up grammar $G_c$ contains context switch labels and lists of follow-up question templates. For example, if the context switch label is INS(SELECT.column0), the corresponding question could be *"How about show column0 too?"*. To ensure generalization, we only induce the grammars from the SPARC training set. Appendix B shows examples of the grammar rules and synthesized utterances.

---

**Algorithm 1** Data synthesis algorithm

1: $\tilde{h} \leftarrow \emptyset$
2: $r_s \leftarrow \text{SAMPLE}(G_s)$
3: $\tilde{u}_0, \tilde{q}_0 \leftarrow \text{RANDASSIGNSLOTS}(d, r_s)$
4: $\tilde{h} += (\tilde{u}_0, \tilde{q}_0)$
5: $\tilde{u}_p, \tilde{q}_p \leftarrow \tilde{u}_0, \tilde{q}_0$
6: **for** $t \leftarrow 1$ to $T$ **do**
7:     **if** $\text{RAND}(0, 1) < 0.2$ **then**
8:         $r_s \leftarrow \text{SAMPLE}(G_s)$
9:         $\tilde{u}_t, \tilde{q}_t \leftarrow \text{RANDASSIGNSLOTS}(d, r_s)$
10:     **else**
11:         $r_c \leftarrow \text{SAMPLE}(G_c)$
12:         **if** $\text{CONSTRAINTCHECK}(r_c, \tilde{q}_p)$ **then**
13:             $\tilde{u}_t, \tilde{q}_t \leftarrow \text{EDITASSIGN}(\tilde{q}_p, r_c)$
14:     $\tilde{h} += (\tilde{u}_t, \tilde{q}_t, r_c)$
15:     $\tilde{u}_p, \tilde{q}_p \leftarrow \tilde{u}_t, \tilde{q}_t$
16: **return** $\tilde{h}$

---

The data synthesis procedure using the two grammars is shown in Algorithm 1. Given a database $d$ and a sampled single-turn question-SQL template, the function RANDASSIGNSLOTS samples values (column names, cell values, and SQL operations) for typed slots in the template and returns the first synthesized question $\tilde{u}_0$ and the corresponding SQL query $\tilde{q}_0$. To generate $T$ follow-up question-SQL pairs, the function CONSTRAINTCHECK$(r_c, \tilde{q}_p)$ checks if the previous query $\tilde{q}_p$ satisfies constraints of the sampled template $r_c$ (e.g. contains its mentioned nonterminal). Finally, EDITASSIGN$(\tilde{q}_p, r_c)$ edits the previous SQL $\tilde{q}_p$ to generate the current follow-up SQL label $\tilde{q}_t$ and samples values for typed slots in the template to generate the corresponding follow-up question $\tilde{u}_t$.

## 3 Experiment Settings

### 3.1 Datasets and Evaluation Metrics

We evaluate SCORE on four popular CSP tasks: SPARC (sequential text-to-SQL), COSQL (conversational text-to-SQL), MWOZ (dialogue state tracking), and SQA (sequential question answering), summarized in Table 1.

**SPARC** (Yu et al., 2019b) [1] is a large collection of sequences of inter-related context-dependent question-SQL pairs. It contains 4.3K questions sequences and 12k+ questions. **COSQL** (Yu et al., 2019a) [2] is a large conversational text-to-SQL corpus, with 3k dialogues, collected under the Wizard-of-Oz (WOZ) setting. We focus on the SQL-grounded dialogue state tracking task which maps user intents into SQL queries if possible given the interaction history. Both SPARC and COSQL cover 200 complex DBs spanning 138 domains.

---

[1] https://yale-lily.github.io/sparc
[2] https://yale-lily.github.io/cosql

**MWoZ** (Budzianowski et al., 2018; Eric et al., 2019) [3] is a corpus of over 10k human-human written task-oriented dialogs created through a WOZ crowdsourcing setting. We focus on the belief state tracking task in MWoZ which maps multi-turn user utterances to slot-value annotations.

**SQA** (Iyyer et al., 2017) [4] is constructed from a subset of WikiTableQuestions (Pasupat & Liang, 2015) by decomposing highly compositional questions into a sequence of simple questions. The task is weakly-supervised because each resulting decomposed question is only annotated with answers as one or more table cells, while the logic program is latent. It has 6,066 question sequences with 17,553 questions in total on 982 unique open-domain tables from Wikipedia.

We adopt the official metrics defined for each of the tasks. For SPARC and COSQL, we report question match accuracy (QM): the exact set match accuracy (Yu et al., 2018b) over SQL templates and interaction match accuracy (IM): the ratio of interactions for which all questions are predicted correctly. For MWoZ, we report joint goal accuracy (JGA) which is similar to the IM accuracy used in SPARC and COSQL. Finally, for SQA, we report denotation QM and IM accuracies.

### 3.2 BASE MODELS AND OTHER BASELINES

For SPARC and COSQL, we use RAT-SQL (Wang et al., 2020) as our base model. Since it is originally developed for single-turn text-to-SQL, we extend it to a multi-turn setting by concatenating current utterances and dialog history (see Section 2.2). Note that RAT-SQL alone, without SCORE, achieves better or comparable results to state-of-the-art models developed for SPARC and COSQL.

For MWoZ, we employ Trippy (Heck et al., 2020). It achieves state-of-the-art performance on MWoZ and uses $BERT_{base}$ to encode user and system utterances and dialog history. We report higher results (around 2%) for Trippy than reported by Heck et al. (2020) since we train it for more epochs (25 vs. 10). To show the improvement of SCORE is not tied to specific base systems, we also experiment with another strong base model SOM-DST (Kim et al., 2020) for MWoZ and follow the same experimental details to train it.

For SQA, we use the weakly-supervised semantic parser proposed by Wang et al. (2019). The model first generates an abstract program given an input question and then instantiates it by searching for alignments between slots in the abstract program and question spans. As it is originally developed for single-turn questions, we extend it to the multi-turn setting in the same way as RAT-SQL.

We report additional implementation details for all base models in Appendix C. In addition to reporting results for all base models with SCORE, we also report original base models results (with BERT and/or ROBERTA) and several other state-of-the-art baselines for each task.

### 3.3 DATASET USAGE IN PRE-TRAINING

In our experiments and ablation study, we train several versions of SCORE with different objectives and datasets: (1) SCORE (MLM): pre-trained on annotated natural questions using MLM. (2) SCORE (CCS+TCS): pre-trained on only synthesized data, which achieves the best results on SParC, CoSQL, and SQA. (3) SCORE (CCS+TCS+MLM): pre-trained on the synthesized data using CCS+TCS and annotated natural questions using MLM.

Furthermore, note that the synthesized data is generated using grammar induced by about 500 examples from only SPARC. Therefore, no COSQL or SQA data are seen in any pre-training steps. For MWoZ, Campagna et al. (2020) study only the dev examples to induce the data synthesis grammar.

### 4 RESULTS AND ANALYSIS

**Overall Results** The results of SPARC and COSQL, MWoZ, and SQA are in Table 2, 3, and 4 respectively. We run each main experiment three times with different random seeds and report the mean. Overall, SCORE gains significant improvements over BERT and ROBERTA on all tasks, achieving state-of-the-art performances on SPARC, COSQL, and MWoZ.

---

[3] https://github.com/budzianowski/multiwoz
[4] http://aka.ms/sqa

| Models | SPARC | | | | COSQL | | | |
| --- | --- | --- | --- | --- | --- | --- | --- | --- |
| | Dev | | Test | | Dev | | Test | |
| | QM | IM | QM | IM | QM | IM | QM | IM |
| SyntaxSQL (Yu et al., 2018a) | 18.5 | 4.3 | 20.2 | 5.2 | - | - | 14.2 | 2.2 |
| GAZP + BERT (Zhong et al., 2020) | 48.9 | 29.7 | 45.9 | 23.5 | 42.0 | 12.3 | 39.7 | 12.8 |
| EditSQL + BERT (Zhang et al., 2019c) | 47.2 | 29.5 | 47.9 | 25.3 | 39.9 | 12.3 | 40.8 | 13.7 |
| IGSQL + BERT | 50.7 | 32.5 | 51.2 | 29.5 | 44.1 | 15.8 | 42.5 | 15.0 |
| $R^2$SQL + BERT | - | - | 55.8 | 30.8 | - | - | 46.8 | 17.0 |
| RAT-SQL + BERT (Wang et al., 2019) | 56.8 | 33.4 | - | - | 48.4 | 19.1 | - | - |
| + ROBERTA | 58.2 | 36.7 | - | - | 50.1 | 19.3 | - | - |
| + SCORE | **62.2** | **42.5** | **62.4** | **38.1** | **52.1** | **22.0** | **51.6** | **21.2** |

Table 2: The SPARC and COSQL accuracy over all questions (QM) and all interactions (IM). The scores of IGSQL + BERT and $R^2$SQL + BERT are from the official leaderboards.

| Models | MWOZ 2.1 |
| --- | --- |
| DST-reader (Gao et al., 2019) | 36.40 |
| TRADE (Wu et al., 2019) | 46.60 |
| DS-DST (Zhang et al., 2019a) | 51.21 |
| SOM-DST (Kim et al., 2020) | 52.57 |
| DS-picklist (Zhang et al., 2019a) | 53.30 |
| TripPy (Heck et al., 2020) | 55.29 |
| SimpleToD (Hosseini-Asl et al., 2020) | 55.72 |
| TripPy (ours) | 58.37 |
| + SCORE | **60.48** |

Table 3: Joint goal accuracies (JGA) on MWOZ 2.1 test set. All models use a BERT-like encoder/GPT.

| Models | SQA | |
| --- | --- | --- |
| | QM | IM |
| Pasupat & Liang (2015) | 33.2 | 7.7 |
| Neelakantan et al. (2017) | 40.2 | 11.8 |
| Iyyer et al. (2017) | 44.7 | 12.8 |
| Sun et al. (2019a) | 45.6 | 13.2 |
| Müller et al. (2019) | 55.1 | 28.1 |
| Herzig et al. (2020) | **67.2** | **40.4** |
| Wang et al. (2019) + RoBERTa | 62.8 | 33.2 |
| Wang et al. (2019) + SCORE | **65.4** | **38.5** |

Table 4: Question (QM) and interaction (IM) accuracy on the SQA test set.

For SPARC and COSQL in Table 2, compared with ROBERTA, SCORE boosts the performance by 4.0% QM / 5.8% IM on SPARC, and 2.0% QM / 2.7% IM on COSQL. This demonstrates the effectiveness of SCORE on contextual semantic parsing tasks. In addition, on MWOZ dialog state tracking task in Table 3, TripPy achieves 60.5% JGA by replacing BERT with SCORE, outperforming the prior state-of-the-art (Hosseini-Asl et al., 2020) by 4.8%. This indicates that dialog state tracking also benefits from SCORE. Finally, SCORE also achieves higher performance than ROBERTA on weakly supervised sequential question answering SQA task. As Table 4 shows, SCORE improves QM by 2.6% and IM by 4.9% over ROBERTA with Wang et al. (2019) as the base model. This demonstrates that the enhanced ability of semantic parsing and context modeling in SCORE is transferable to denotation-based CSP tasks.

**What is the effect of each pre-training objective?** Table 5 shows an ablation study on different pre-training objectives. We find that the best SCORE results are achieved by pre-training on only synthesized data (CCS+TCS) without any natural questions (MLM) on SPARC,

| Learning Objective | SPARC | COSQL | MWOZ | SQA |
| --- | --- | --- | --- | --- |
| MLM only | 37.0(+0.3) | 20.3(+1.0) | 59.47(+1.10) | 34.7(+1.5) |
| CCS only | 41.3(+4.6) | 21.2(+1.9) | 59.32(+0.95) | 32.7(-0.5) |
| CCS+TCS | **42.5(+5.8)** | **22.0(+2.7)** | - | **38.5(+5.3)** |
| CCS+TCS+MLM | 38.6(+1.9) | 21.7(+2.4) | **60.48(+2.11)** | 33.7(+0.5) |

Table 5: The effect of SCORE pre-training objectives. Improvements are shown in the parentheses.

COSQL, and SQA but not on MWOZ. By adding MLM to CCS+TCS (CCS+TCS vs. CCS+TCS+MLM), MLM actually hurts the performance (-3.9% on SParC, -0.3% on CoSQL, and -4.4% on SQA) while increases for MWOZ. One possible reason is that questions in MWOZ are more diverse in language but less compositional while semantic compositionality and turn changes are more important in the other three CSP tasks. Also, the synthesized data used to pre-train SCORE for SPARC and COSQL is generated by the grammar induced by SPARC, which might overfit to SPARC. In addition, SCORE pre-trained with only MLM loss improves the performance ( 1.0%) but not as large as CCS+TCS (+5.5% on SPARC, +1.7% on COSQL, and +3.4% on SQA). Finally, we test the effectiveness of TCS on SPARC, COSQL, and SQA by adding TCS to CCS (CCS only vs.

CCS+TCS), SCORE gains improvements of 1.2% on SPARC and 0.8% on COSQL, and 4.4% on SQA.

**Does SCORE improve question match accuracy on individual turns?**  Table 6 shows detailed results of SCORE's question accuracy for individual conversation turns on the SPARC dev set. SCORE provides a significant improvement for every conversation turn except the first (in which the task is more similar to single-turn semantic parsing). COSQL and SQA exhibit similar behavior and are presented in Appendix A.

|  | QM | Q1 | Q2 | Q3 | Q4 |
|---|---|---|---|---|---|
| RAT-SQL + BERT | 56.8 | **71.1** | 53.6 | 47.8 | 31.8 |
| +RoBERTa | 58.2 | 68.7 | 58.5 | 48.9 | 35.2 |
| + SCORE | **62.2** | 70.6 | **63.5** | **52.6** | **45.5** |

Table 6: Detailed results on the dev set of SPARC. $Q_i$ is the accuracy of the $i^{th}$ conversation question.

**What if we use the synthesized data to simply augment the training data?**  To answer this, we compare the results of the base models trained with or without the synthesized data on COSQL and MWOZ. As shown in Table 7, the extra synthetic data does not significantly improve the performance, indicating that directly augmenting the synthetic data to the training set is not effective. The similar findings are reported in many recent work (Zhang et al., 2019c; Herzig et al., 2020; Campagna et al., 2020; Zhong et al., 2020). In contrast, pre-training on the synthesized data with our objectives improves the performance on the downstream tasks.

|  | COSQL | MWOZ |
|---|---|---|
| no syn | 48.4 | 58.37 |
| with syn | 48.6 | 58.45 |

Table 7: Effect of synthetic data as training data augmentation.

**How general is SCORE and its synthetic grammar?**  For generalization in task settings, we have shown that the pre-training strategy of SCORE can improve the performance over different CSP tasks including semantic parsing (SPARC and COSQL), dialog state tracking (MWOZ), and weakly supervised table question answering (SQA). In addition, we demonstrate the effectiveness of SCORE on different *base models*. To this end, we experiment with a different base model SOM-DST for MWOZ. As shown in Table 8, SCORE can still improve the performance with a different base model on MWOZ (SOM-DST+BERT vs. SOM-DST+SCORE on syn. MWOZ).

|  | MWOZ |
|---|---|
| SOM-DST + BERT | 52.57 |
| + SCORE on syn. text-to-SQL | 53.57 |
| + SCORE on syn. MWOZ | 54.61 |

Table 8: Performance of SCORE pre-trained on different synthesized data on MWOZ.

To demonstrate the generalization in synthetic grammar and data, as shown in Table 2 and 4, SCORE (TCS+CCS) is pre-trained on the synthesized data of the grammar induced from SPARC *only*, and it still improves the performance on COSQL (+2.7%) and SQA (+4.9%) where *no* any CoSQL and SQA annotated data is seen in any pre-training steps. Moreover, in Table 8 we show that SCORE pre-trained on the text-to-SQL synthesized data could also surprisingly improve the performance on MWOZ. We expect that higher performance could be achieved with SCORE pre-trained on task-specific synthesized data. Finally, our pre-training approach can be applied to *any* existing LMs including larger seq2seq LMs (e.g., BART (Lewis et al., 2020), T5 (Raffel et al., 2020)).

**Can SCORE deliver more value when in-domain data is limited (e.g., in a low-resource setting)?**  We want to answer this question similar to experiments other investigations of LMs as few-shot learners (Wu et al., 2020; Brown et al., 2020; Schick & Schütze, 2020). To this end, we compare ROBERTA and SCORE under a few-shot setting on SQA when only 10% of training data is available. We choose SQA because its annotation is most different from the synthetic text-to-SQL dataset we use for pretraining. Table 9 demonstrates that SCORE delivers even larger improvements compared to the ROBERTA baseline when only 10% training data is available (3.8% vs 2.6%).

|  | QM | IM |
|---|---|---|
| RoBERTa | 53.3 | 21.2 |
| SCORE | **57.1** | **26.1** |

Table 9: Performance of SCORE on 10% training data of SQA.

## 5 RELATED WORK

**Conversational Semantic Parsing**  Conversational semantic parsing is one of the most important research topics in conversational AI and has been studied in different settings including task-oriented dialogues, question answering, and text-to-SQL. Task-oriented dialog systems (Henderson et al.,

2014; Wen et al., 2016; Mrkšić et al., 2017; Budzianowski et al., 2018) aim to help users accomplish a specific task (e.g. flight booking) and often pre-define slot templates grounded in a domain-specific ontology. In comparison, several other datasets were recently introduced for cross-domain conversational text-to-SQL tasks (SPARC and COSQL (Yu et al., 2019a;b)) and sequential questions answers over tables (Iyyer et al., 2017). While the previous work has achieved significant progress in different datasets separately, to the best of our knowledge, we are the first to study four different CSP tasks together (sequential text-to-SQL, conversational text-to-SQL, dialog state tracking, and weakly-supervised sequential question answering) by addressing the shared key challenge of learning representations in pre-trained language models that capture the alignment between the dialogue flow and the structural context.

**Conversational Language Model Pre-training**    Several recent efforts have demonstrated the value of adapting pre-trained LMs to specific tasks using different pre-training objectives, e.g., summarization (Zhang et al., 2019b), knowledge inference (Sun et al., 2019b; Liu et al., 2019a), etc. Closest to our work is adapting pre-trained LMs for open-domain chit-chat models and for tabular data representation. The former focuses on improving response generation on open-ended dialogues by adding a pre-training step on open-domain conversations data, such as Reddit data (Zhang et al., 2020; Henderson et al., 2019). For example, Wu et al. (2020) introduced ToD-BERT, a pre-trained language model combining 9 high-quality human-human task-oriented dialogue datasets to conduct language model and response selection pre-training. However, they use language modeling training objectives over free-form text and therefore have limited ability to represent structural data. The latter has focused on improving language model pre-training for encoding tabular data (Yin et al., 2020; Herzig et al., 2020), but they focus on the single turn semantic parsing setting. Our approach is different from previous work because we address the challenge of conversational semantic parsing tasks by learning pretrained representation for both the multi-turn dynamics of the dialog and the relation between the unstructured language utterance and the structured ontology. Furthermore, our pre-training approach is much more data-efficient than prior LM pre-training work and saves a lot of time and computing resources (Appendix D for more details). Our pre-training step can be done within only one day using 8 V100 GPUs.

**Using Synthesized Data for Semantic Parsing**    Synthesized data has been frequently used in semantic parsing to alleviate the challenge of labeled data scarcity. For example, Wang et al. (2015) proposed a method for training semantic parsers in new domains by generating logical forms and canonical utterances and then paraphrasing the canonical utterances via crowd-sourcing. Similar approaches were used to train semantic parsers in other domains and settings (Zhong et al., 2017; Su et al., 2017; Cheng et al., 2018; Shah et al., 2018). Another line of work has proposed using synthesized data to adapt single turn semantic parsing models to new domains (Jia & Liang, 2016; Yoo et al., 2018; Campagna et al., 2019) and task-oriented dialogues (Campagna et al., 2020). However, they reported that combining synthetic data and the supervised data does not yield significant improvements, consistent with results by Herzig et al. (2020). By contrast, we introduce a new data synthesize procedure for conversational text-to-SQL dialogues and use it in a different way by pretraining language models to induce better representations for many CSP tasks. Our synthesized data can be easily generated without human involvement and the pre-trained models add value to different tasks simultaneously.

## 6    CONCLUSION

We presented SCORE a new pre-training approach for conversational semantic parsing. The training objectives of SCORE aim to induce natural language representations that capture the multi-turn dynamics, compositional semantic of the target language, and the references to the structural ontology appearing in the dialog. SCORE can be used with many semantic parsing models as a drop-in replacement for general pretrained LMs. We demonstrated SCORE effectiveness by using it as a feature representation encoder with strong baseline models for a wide range of CSP tasks. In particular, our empirical results on four different CSP tasks demonstrated that SCORE can be used to significantly improve the performance of existing strong baseline models by simply replacing an existing pre-trained LM with our SCORE pre-trained model. Furthermore, we are able to achieve state-of-the-art results on three of these tasks. We hope SCORE will encourage further exploration of the benefits and limitations of pre-training approaches for CSP systems.

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

## A   DETAILED RESULTS

|  | QM | IM | Q1 | Q2 | Q3 | Q4 | Q5 |
|---|---|---|---|---|---|---|---|
| RAT-SQL + BERT | 48.4 | 19.1 | 54.6 | 48.4 | **47.5** | 43.9 | 31.0 |
| +RoBERTa | 50.1 | 19.3 | 59.7 | 50.9 | 46.3 | 46.5 | **32.4** |
| + SCORE | **52.1** | **22.0** | **60.8** | **53.0** | 47.5 | **49.1** | **32.4** |

Table 10: Detailed results of COSQL on the dev set. $Q_i$ is the accuracy of the $i^{th}$ question in the conversation.

|  | QM | IM | Q1 | Q2 | Q3 |
|---|---|---|---|---|---|
| Wang et al. (2019) | 51.0 | 22.0 | 68.3 | 48.0 | 38.5 |
| +RoBERTa | 62.8 | 33.2 | 77.2 | 61.7 | 52.1 |
| +SCORE | **65.4** | **38.5** | **78.4** | **65.3** | **55.1** |
| Few-Shot (10% training data) |  |  |  |  |  |
| Wang et al. (2019) |  |  |  |  |  |
| +RoBERTa | 53.3 | 21.2 | 71.0 | 52.5 | 36.6 |
| +SCORE | **57.1** | **26.7** | **74.6** | **56.7** | **40.7** |

Table 11: Detailed results of SQA on the test set. $Q_i$ is the accuracy of the $i^{th}$ question in the conversation.

## B   SYNTHESIZED EXAMPLES & TEMPLATES

Table 12 shows an example of the synthesized question-SQL pairs and their corresponding templates in our grammars.

| Turn # | Question-SQL Template | Synthesized Question-SQL |
|---|---|---|
| 1 | "Find the number of TABLE0 with COLUMN0 OP0 VALUE0" SELECT COUNT(*) ORDER BY COLUMN0 OP0 VALUE0 | "Find the number of football team with team hometown is not murrieta, california?" SELECT COUNT(*) WHERE TEAM_HOMETOWN != "MURRIETA, CALIFORNIA" |
| 2 | "Can you give me their COLUMN1?" TCS:      REPLACE(SELECT.COLUMN0), DEL(SELECT.AGG) | "Can you give me their football team player?" SELECT FOOTBALL_TEAM_PLAYER WHERE TEAM_HOMETOWN != "MURRIETA, CALIFORNIA" |
| 3 | "How about only show those with AS0 COLUMN2?" TCS: ADD(ORDERBY_AS0.COLUMN2) | "How about only show those with the largest age?" SELECT FOOTBALL_TEAM_PLAYER WHERE TEAM_HOMETOWN != "MURRIETA, CALIFORNIA" ORDER BY AGE DESC LIMIT 1 |
| 4 | "AS1?" TCS: REPLACE(ORDERBY_AS1.COLUMN2) | "The smallest?" SELECT FOOTBALL_TEAM_PLAYER WHERE TEAM_HOMETOWN != "MURRIETA, CALIFORNIA" ORDER BY AGE AS LIMIT 1 |

Table 12: An example of synthetic conversational text-to-SQL data.

## C   IMPLEMENTATION DETAILS

### C.1   SCORE

For pre-training SCORE on synthesized text-to-SQL data, we use ROBERTA $_{large}$ and pre-train it with batch size 12, gradient accumulation step 2, and maximum length 248. We use a learning rate $1e$-5 and gradually reduce the learning rate without a warm-up period using Adam (Kingma & Ba, 2014) with epsilon $1e$-8. BERT$_{base}$ is used in pre-training SCORE on synthesized MWOZ data because it contains longer conversations. We set the maximum length to 512 and batch size 24. All SCORE are pre-trained for 30 epochs, which usually take less than half a day on 8 V100 GPUs.

We experimented with SCORE pre-trained for 5, 10, and 30 epochs and found that most of the best downstream performances occur when base systems incorporate with SCORE pre-trained for less than 10 epochs. Our implementation is based on the Transformers library (Wolf et al., 2019).

## C.2   BASE MODELS

**RAT-SQL:** For a fair comparison, all RAT-SQL experiments are trained for 40k steps. We adopt the same hyperparameters as Shaw et al. (2018) except for learning rates. We find that learning rates of $1e$-4 and $1e$-5 for RAT and BERT respectively produce more stable results.

**TripPy:** We use the same hyperparameters for training TripPy on MWOZ as in (Heck et al., 2020) except we train it for 25 epochs (as opposed to 10 epochs as reported in (Heck et al., 2020)). When we train TripPy for 25 epochs, we get a new result that is higher (around 2%) than the one reported in (Heck et al., 2020). Similarly, when we train TripPy with SCORE, we train it for 25 epochs.

**SOM-DST:** We use the same hyperparameters from Kim et al. (2020) for all SOM-DST experiments on MWOZ.

**Wang et al. (2019):** We use the same hyperparameters from Wang et al. (2019) for SQA experiments. Note that Herzig et al. (2020) outperform Wang et al. (2019) on SQA because (1) they don't generate logic forms but select table cells and applying aggregation operators. Wang et al. (2019) generate latent programs, yet the grammar of the latent program can only cover 87% questions. (2) They reduce the search space by reusing the previous question answer. We choose Wang et al. (2019) as our base model because generating symbolic programs has many practical advantages (even at a cost of around 1% accuracy drop), such as showing interpretable reasoning steps, enabling formal reasoning, and operationalization without GPU/TPU accelerators.

## D   PRE-TRAINING COST

We test the performance of SCORE with respect to the number of pre-training epochs. Figure 3 shows that the best performance of the downstream tasks is usually achieved in early epochs, more specifically 5 for SPARC and COSQL and 15 for MWOZ. Longer pre-training time does not improve or even hurts the performance. One possible reason is that longer pre-training makes SCORE overfit to the synthesized data whose utterances are unnatural.

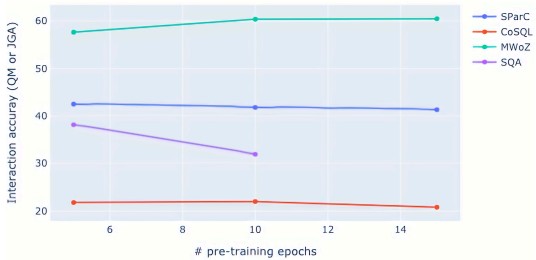

Figure 3: The effect of pre-training time.

As for the data, as shown in Table 5, even if SCORE is pre-trained with only a relatively small amount of synthesized data (without the MLM loss), most of the tasks can achieve much higher performances. With a relatively smaller training corpus and shorter training time compared to other pre-trained language models, SCORE is efficient in time and data.

## E   ADDITIONAL RESULTS

**Effect of TCS**   We ran the TCS only experiment on SPARC, and will add TCS only results (including for other tasks) to Table 5 in the final version. SCORE (TCS only) outperforms RoBERTa by 2.4% so far (note: training is still going on) on SPARC (39.1% vs. 36.7%). Also, as discussed in Section 4, we also provide a secondary evidence by testing the effectiveness of TCS on SPARC, COSQL, and SQA by adding TCS to CCS (CCS only vs. CCS+TCS), SCORE (with TCS) gains improvements of 1.2% on SPARC and 0.8% on COSQL, and 4.4% on SQA.

**Incorporating Additional Examples Used in Synthetic Grammar Induction**   As we mentioned in Section 2.3, we used about 500 examples from SPARC to induce the grammar for data synthesis in pre-training. For a fair comparison, we also report the results of incorporating the additional SPARC examples in COSQL and SQA. More specifically, we directly concatenate the additional SPARC

examples to CoSQL training set, and train RAT-SQL+RoBERTa on it, which slightly improves the performance (19.6% vs. 19.3%) but not as large as SCoRE (22.0% vs. 19.3%).' Also, because SQA is weakly-supervised sequential question answering, which differs from SParC, we first fine-tune RoBERTa on the additional SParC examples using CCS, and then apply it to SQA. In this way, the RoBERTa trained with additional SParC examples achieves a similar performance as the original one (62.7% vs 62.8%).

**Performance Comparison with ToD-BERT**    ToD-BERT is pre-trained on human-annotated questions with both MLM and response contrastive objectives. To compare TOD-BERT with SCoRE, we ran experiments of RAT-SQL + ToD-BERT on SParC. SCoRE (62.2%) outperforms ToD-BERT (54.6%) by 7.6%.

**Comparison with Finetuning Larger Language Models**    Based on our experiments and other published results, we didn't find existing larger LMs (BART (Lewis et al., 2020), T5 (Raffel et al., 2020), GPT-2 (Radford et al., 2019)) outperform custom models + BERT on CSP tasks. Our evidence is based on Spider (Yu et al., 2018b), which is the single-turn version of SParC and CoSQL. For T5, Shaw et al. (2020) applied T5 as seq2seq to Spider, and compared with RAT-SQL + BERT-Large, T5-Base performs much worse (57.1% vs. 69.6%), and T5-3B improves only 0.3, but it is 6 times larger. Moreover, for Bart, we have performed experiments on Spider and we found that BART cannot outperform custom models + BERT: RAT-SQL + BERT 69.7%, RAT-SQL + BART encoder 67.8%, BART encoder + decoder (406M, as a seq2seq task) 62.4%. In Rubin & Berant (2020), BART didn't outperform BERT either. As for GPT-2, Wu et al. (2020) and Hosseini-Asl et al. (2020) found it does not outperform BERT on MWoZ.

# F    TASK-ORIENTED DIALOGUE DATASETS

| Name | # Dialogue | # Utterance | Avg. Turn | # Domain |
|---|---|---|---|---|
| MetaLWOZ (Lee et al., 2019) | 37,884 | 432,036 | 11.4 | 47 |
| Schema (Rastogi et al., 2019) | 22,825 | 463,284 | 20.3 | 17 |
| Taskmaster (Byrne et al., 2019) | 13,215 | 303,066 | 22.9 | 6 |
| MWOZ (Budzianowski et al., 2018) | 10,420 | 71,410 | 6.9 | 7 |
| MSR-E2E (Li et al., 2018) | 10,087 | 74,686 | 7.4 | 3 |
| SMD (Eric and Manning, 2017) | 3,031 | 15,928 | 5.3 | 3 |
| Frames (Asri et al., 2017) | 1,369 | 19,986 | 14.6 | 3 |
| WOZ (Mrkšić et al., 2016) | 1,200 | 5,012 | 4.2 | 1 |
| CamRest676 (Wen et al., 2016) | 676 | 2,744 | 4.1 | 1 |

Figure 4: Data statistics of human-annotated task-oriented dialogue datasets used in Wu et al. (2020).

