# OpenReview forum: "SCoRe: Pre-Training for Context Representation in Conversational Semantic Parsing"
_ICLR.cc/2021/Conference — ICLR 2021 Poster_

### Official Review · AnonReviewer4 · 2020-10-27
**Well written paper introducing a pre-training objective specifically for semantic parsing.**

**Rating:** 7
**Confidence:** 4

**Review:**

Summary:
This paper introduces a semantic-parsing specific pretraining objective. The authors argue that general pre-training methods such as BERT do not have enough inductive bias for semantic parsing.
Since a lot of data doesn’t exist for semantic parsing datasets,  the authors use synthetic data to better adapt to the ontology of the task. They use this pre-trained model on 4 semantic parsing tasks and show that their pre-training is indeed helping the SOTA models by establishing new SOTA for 3 of the 4 datasets.

Reasons for score:
This is a clearly written paper with the objective of making conversational semantic parsing better. The authors present a reasonable idea, do extensive experiments to show its merit on variety of tasks. The pre-trained checkpoints released by this paper will be useful to the community as a whole.

Pros:
1. 2 new objective functions (TCS and CCS) specific to the task of semantic parsing. Authors clearly show that both those objectives help downstream tasks
2. The paper was easy to follow with clear objectives.
3. Strong performance of their pre-trained model which improves over previous SOTA
4. Well done ablation studies and analysis.

Cons:
1. Data synthesis steps are ad-hoc. Why were only 435k dialogues synthesized? It would be great to have a more detailed study of how the synthetic data looks, what is the effect on model performance.
2. Due to synthetic data the method is not very general. Training larger models with more data will be non trivial to do.
3. While the authors chose SOTA baselines for their task, stronger general pre-trained models (BART, T5 etc) might beat this method easily.

Please address and clarify the cons above

Typos/Areas for improvement:
1. Citations in table 3
2. Mistake in Table 1 Multiwoz2.1 is multi domain.
3. The number of utterances are not clear: 3.1 -> vanilla multiwoz2.1 does not have over 100k task oriented dialogues
4. 3.1 SQA description not clear.
5. In section 2.2 it would be great to have a clearer description of the steps for pre-training, it's hard to tease out the exact steps taken
6. For table 7, what happens when you use the synthetic tuning data to first train and then fine-tune on the original task.
7. For table 8: More studies on the generalizability would be good
    1. If you simply drop in the network to something small like geo-query, does it help a general seq2seq model?
    2. Beyond semantic parsing tasks - does this help in sentence classification too?

---

> ### Author Response · Authors · 2020-11-14
> **Author Response**
>
> Thanks for your feedback!
>
> **Data synthesis steps**
>
> The reason why we synthesized 435k dialogues is because there are about 400k tables from multiple online resources after filtering and cleaning and then we synthesize about one dialog for each table. We will add more details on data synthesis to the rebuttal version. In Appendix, Table 11 shows an example of synthesized data and corresponding templates in our grammars.
>
> **Generalizability concern due to the synthesized data**
>
> Thanks for raising this point! We also considered this problem when we were working on this project. To show our pre-training method *is actually general* (not tied with a grammar induced from a specific data), we only study a very small set of SParC dataset in order to induce the grammar generation rules. SCoRe is pre-trained on the data synthesized by the grammar (induced from a small subset of SParC), and could still improve the performance on CoSQL and SQA. Also, CSP questions are more compositional than other text since they can be mapped into some formal representation. Data synthesis using induced grammar is widely used to enlarge data size.
>
> **SOTA base systems vs. “stronger” general pre-trained models**
>
> We agree that these kinds of experiments would be interesting to have. However, the main focus of our paper is to show the effectiveness of the proposed CSP pre-training method for a practical-sized pretrained LM. We are not aware of any published results using large LMs such as BART/T5 for conversational text-to-SQL, as they are challenging to fine-tune and deploy and require access to computational resources that are not readily available. Please also check out our response to a similar question by Reviewer 5.
>
> We believe that no matter what base systems or language models will be used, the pre-training approach is likely to improve their performance on CSP tasks (either incorporated with base systems or directly pre-train the language models), but this will require additional experiments to verify. In Section 4, we show the *effectiveness of SCoRe improving multiple independent base systems*.
>
> **Areas for improvement**
>
> We will consider adding more studies on general seq2seq models for small single-turn datasets and sentence classification tasks, yet SCoRe is designed for CSP tasks.
>
>
> **Typos and Clarity**
>
> Thanks for catching these! In our rebuttal revision, we will add citations in Table 3, correct statistics about MultiWOZ 2.1, add more description of SQA, add more details on pre-training steps.

---

### Official Review · AnonReviewer1 · 2020-10-27
**Interesting work, would benefit from improved discusison**

**Rating:** 7
**Confidence:** 3

**Review:**

Overview: the authors propose a new pre-training method for grounding LM representations in both structural/schematic information and also dialogue context. To explore the efficacy of this method, they experiment in 4 different conversational semantic parsing tasks, which are each different enough to demonstrate the usefuleness of their approach.

Contribution: modified pre-training method

The good: My overall impression is that this work makes sense, the paper is clearly written and flows just fine, and that the authors demonstrated the efficacy of their proposed method. The fact that the 4 baselines are different enough makes your claim convincing.

The bad: I feel that this paper is really lacking a driving motivation and the cohesive story is a bit weak/lacking. For example, I felt the Related Works section was very superficial, rather than contributing. Shouldn't the discussion/conversation be more about contextualization methods?  You are not the first to try contextualizing schema elements, for example. I think the paper could be significantly improved if the conversation is shifted towards discussion of contextualization, rather than "CSP is a task. People use pre-training. You can generate some data and use it". You have an interesting story in your hands, but you are not using it! How is your contextualization method different from other attempts? In what ways is it better, in what ways is it worse? That said, I still think the paper has good and interesting work, and I do think it should be accepted, but I would really encourage the authors to consider making some changes on this note because it will make the work much more interesting, thought provoking, and impactful.

Small things:
* unless i click on the box to jump to Table 1 on the PDF, it took me some time to find Table 1 when I printed it out. Is there anyway to move this closer to where you point to it, or sepperate it from the figure below?

Clarifying question:
* With regards to encoding the Turn Contextual Switch -- are you encoding the changes from user to agent, or just the changes from user to user, or both? Is it always user to user query changes, because some datasets have no system response? This is the only section in the paper that I found to be lacking a bit, and maybe a sentence or two more could add clarification for me.

Overall:
Good work :-)

---

> ### Author Response · Authors · 2020-11-14
> **Author Response**
>
> Thanks for your feedback! We will update our paper in our rebuttal revision to make a more compelling story with comparisons with other contextualization methods as follows. We are happy to incorporate any future comments.
>
> **Our motivation and story**
>
> We will work on the introduction and related work sections to clarify the motivation and better position the work w.r.t other contextualization methods. To summarize our motivation and story, we observe that the shared key challenge in several different CSP tasks (dialog state tracking, context-dependent semantic parsing, SQL-grounded state tracking, sequential question answering) is how to jointly represent the natural language utterances and underlying structured ontology while taking into consideration the multi-turn dynamics of the dialog. Questions in these CSP tasks are more compositional than other text since they can be mapped into formal representations, and therefore we would like to design a more effective pre-training approach to inject this kind of compositional bias in LMs. Unlike other prior work on data synthesis using an induced grammar to enlarge data size, we propose to pre-train LMs on synthesized data and design CSP related pre-training objectives (CCS and TCS) that creates representations to improve several CSP tasks.
>
> **Comparisons with related work on other contextualization methods**
>
> We will update the related work section with an improved discussion on other contextualization methods and highlighted our contributions compared to previous work:
> 1. While the previous work has achieved significant progress in different datasets separately, to the best of our knowledge, we are the first to study four different CSP tasks together (sequential text-to-SQL, conversational text-to-SQL, dialog state tracking, and weakly-supervised sequential question answering) by addressing the shared key challenge of learning representations in pre-trained language models that capture the alignment between the dialogue flow and the structural context.
> 2. Our approach is different from previous work because we address the challenge of conversational semantic parsing tasks by learning pre-trained representation for both the multi-turn dynamics of the dialog and the relation between the unstructured language utterance and the structured ontology.
> 3. We introduce a new data synthesize procedure for conversational text-to-SQL dialogues and use it in a different way by pretraining language models to induce better representations for many CSP tasks.
> 4. Our pre-training approach is much more data-efficient than prior LM pre-training work and saves a lot of time and computing resources. Our pre-training step can be done within only one day using 8 V100 GPUs.
>
> **Clarification on TCS**
>
> Turn Contextual Switch encodes changes between different turns of user queries (the system response is not involved here) since we assume that most turn contextual shifts are from the user.
>
> **Paper Format**
>
> Thanks for pointing this out! Table 1 is pointed in multiple places, so it was hard to move it closer to all the references. We will separate it from the figure below and put it closer to the position where the first reference occurs (like right before Section 2) in the rebuttal version.

---

### Official Review · AnonReviewer3 · 2020-10-27
**Official Blind Review #3 (Edited post author response period)**

**Rating:** 7
**Confidence:** 4

**Review:**

This paper proposes a pre-training approach to improve the performance in conversational semantic parsing.  The idea is to use the training data to learn how to generate contextual representations by combining the now commonly used masked language modelling pretraining objective (MLM) with two additional objectives, named column contextual semantics and turn contextual switch. Furthermore, additional synthetic data was generated.

I found the paper easy to follow and the results convincing on the whole. 4 datasets and different parsers were used and in each case the proposed method improved the results, and in 3 out of 4 a new SOTA was reached.

However, it is important to note that the objective propose needs labeled training data in order to learn the alignment between the utterances and the queries. Thus they allow us to exploit labeled data for other versions of the task, rather than exploit unlabelled data for the semantic parsing. Therefore, while I like the paper, I think a key comparison missing is to compare against training existing models using combinations of the datasets considered, to allow the models access to the same training data. This could be easy incases where the output is of the same form, e.g. SQL for SPARC and COSQL, or it could be multi-task training to combine it with the datasets fo the other two tasks. As things stand, the approach proposed indeed brings benefits, but it is only compared against methods using no additional labeled data beyond what is built for the task at hand. Having said this, I am not arguing that training data concatenation or multi-task training will work better; but I think such a comparison is needed.

Beyond this, the other aspect of the paper that needs to be improved is the technical clarity. While I found the intuitive descriptions of equations 2 and 3 easy to understand, the equations themselves were not. In both cases the formal representation q (e.g. the SQL query) while it is needed to calculate the objective, it is not in the equations. Thus it would be very easy to have it interpreted differently and thus lead to ambiguity and people not being able to reproduce it. This is particularly important as how one does the decomposition of q is likely to matter to the results.

Post-author response: I appreciate that the extra experiment I asked for was conducted and the equations were fixed. While I think Review5 raised some interesting discussion points which should be included in the final version, I still think the paper has merit, even if larger-scale pre-training would have improved the results. Thus I raised my score to 7.

---

> ### Author Response · Authors · 2020-11-14
> **Author Response**
>
> Thanks for your feedback!
>
> **Using labeled data in pre-training method**
>
> Thanks a lot for your thoughtful comments and suggestions. We would like to first clarify that our pre-training approach *doesn’t use all labeled training data in the four datasets but only a small set of SParC dataset* (~500 examples, to manually induce the data synthesis grammar). Then we apply the induced grammar to synthesize data for pretraining using CCS and TCS to learn the alignment of utterances and the queries. Therefore,
> 1. In our SParC experiments, both SCoRe (only the ~500 examples) and baseline models only use SParC training data;
> 2. For CoSQL and SQA experiments, SCoRe pre-training doesn’t have access to any CoSQL and SQA labeled data;
> 3. To show our pre-training method is not tied with a grammar induced from dataset-specific examples, as shown in Table 2 and 4, SCoRe could still improve the performance on CoSQL and SQA even though its data synthesis grammar is only induced from SParC.
>
> We will make this clearer in our rebuttal revision. Please also check out our response to a similar question of Reviewer 5.
>
> Additional comparisons against data concatenation or multi-task training (e.g., training a single model on all concatenated labeled data in the four tasks) would be very interesting, yet since we only use a small amount of labeled SParC data in our SCoRE pretraining, using an equivalent amount of data is unlikely to significantly help.
>
>
> **Technical Clarity**
>
> Thanks for this suggestion! We will update equations 2 and 3  (e.g.) by including the formal representation q in our rebuttal revision.

---

> > ### Comment · AnonReviewer3 · 2020-11-17
> > **Response to response**
> >
> > I appreciate the response. I think though that the comparison I suggested against data concatenation or multi-task training is something needed. Sure the amount of training data used is small, but could still improve the results but differences obtained by adding SCoRe are not huge either.

---

> > > ### Author Response · Authors · 2020-11-20
> > > **Thanks for your response**
> > >
> > > Thanks for your response! We are working on the experiments you suggested and try to get at least some of them done by the rebuttal deadline.

---

> > > ### Author Response · Authors · 2020-11-25
> > > **Author Response**
> > >
> > > We used about 500 examples from SParC to induce the grammar for data synthesis in pre-training. For a fair comparison, we have incorporated the additional SParC examples in CoSQL and SQA. We found this does not significantly improve the performance on CoSQL and SQA compared with SCoRe. Please check Appendix E for more details.

---

### Official Review · AnonReviewer2 · 2020-11-03
**TCS needs further investigation**

**Rating:** 6
**Confidence:** 4

**Review:**

The paper proposes to pretrain contextual semantic parsing models on synthesized data with two new training objectives: Column Contextual Semantics (CCS) and Turn Contextual Switch (TCS). The CCS objective predicts correct database operations based on corresponding columns in tables. The TCS aims to predict the labels of conversational turn switch patterns categorized based on differences in meaning representations between dialogue turns. The synthetic data is generated by apply two utterance-SQL generation grammars. They show that the new approach significantly outperforms te baselines on Sparc, CoSQL, and MultiWOZ.

My decision is just between marginally accepted and marginally rejected. I like the idea of pretraining with CCS and the empirical results show that the proposed approach outperforms all baseline systems on three out of four benchmark datasets. However, my major concern is the usefulness of TCS.

Pros:

1. The paper finds out that CCS and synthetic data works in pre-training, despite the prior work finds synthetic data is not useful in a standard supervised setting.

2. Overall, the paper is well written. I can easily follow the technical details of the proposed methods.

3. This paper provides comprehensive experiments to justify the key contributions of this paper. The ablation study helps understand which technique works on the selected datasets.

Cons:

1. The usefulness of TCS objective needs further justification. I suggest adding experiments of TCS only to Table 6. The TCS objective needs further investigation to understand in which cases it works.

2. It is desirable to have an ablation study to investigate how effective is each grammar (with/without follow-up context-free grammar) for pretraining.

Questions:
1. Could you explain why the proposed method works worse than herzig et al. (2020b) on SOA?

2. Why does the system compare with the baselines also trained on 10\% training data of SOA?

---

> ### Author Response · Authors · 2020-11-14
> **Author Response**
>
> Thanks for your feedback!
>
> **Usefulness of TCS objective**
>
> Thanks for the suggestion! Yes, we'll do that. Also, you can get secondary evidence for it by comparing TCS+CCS with CCS only that are already in Table 6 (the performance is improved on all the three tasks especially SQA by adding TCS.). We'll run the TCS-only experiment during the rebuttal period and will update the PDF (likely for one dataset during the rebuttal period and all others for the camera-ready).
>
> **Effectiveness of each grammar**
>
> To clarify this suggestion, did you mean template rules or the whole grammar? For each interpretation:
> 1. Ablation by individual template rules. The set of ablations will be very large, hard to interpret and the computational expense will be prohibitive.
> 2. Ablation by grammars. This type of ablation will not be informative. If we don't use the follow-up grammar, we cannot generate follow-up conversations, which (a) reduces the problem to single-turn semantic parsing, and (b) effectively drops CCS+TCS in pre-training entirely. If we don't use context-independent grammar, we cannot generate the first questions in conversations.
>
> **Comparison with Herzig et al. (2020) on SQA**
>
> Herzig et al. (2020) outperform Wang et al. (2019) on SQA because (1) they don't generate logic forms but answer questions on tables by selecting table cells and applying aggregation operators. Wang et al. (2019) generate latent programs to answer questions yet the grammar of the latent program can only cover a subset of questions. (2) They reduce the search space by reusing the answer to the previous question to answer the current question. However, we showed that SCoRE can improve Wang et al. (2019) with RoBERTa and achieve a performance of 65.4% very close to SOTA of 67.2%. We will still try to improve our performance, but the current experimental results have already shown the effectiveness of SCoRE.
> Furthermore, we also believe that generating symbolic programs has many practical advantages (even at a cost of ~1% accuracy drop), such as showing interpretable reasoning steps, enabling formal reasoning, and operationalization without GPU/TPU accelerators. We will add this to our rebuttal revision.
>
> **Few-shot learning with 10% training data on SQA**
>
> In this experiment, we want to answer the research question of "Can SCoRE deliver more value when in-domain data is limited (e.g., in a low-resource setting)?" This is similar to experiments done in ToD-BERT paper and other investigations of LMs as few-shot learners. We choose SQA for this experiment because the SQA annotation is most different from the synthetic text-to-SQL dataset we use for pretraining SCoRE. In Table 4, the setting of 10% SQA training data demonstrates that SCoRE delivers more benefits compared to the RoBERTa baseline under a low-resource setting where only a small amount of task-specific annotations are available. When all training data is used, SCoRE of 65.4% outperforms RoBERTa of 62.8% by 2.6% QM accuracy, and when only 10% training data is available to both models, the improvement (57.1% - 53.3% = 3.8%) is even larger. We will add this analysis as a separate paragraph in Section 4 in our rebuttal revision.

---

### Official Review · AnonReviewer5 · 2020-11-04
**Review (Edited after comments)**

**Rating:** 4
**Confidence:** 5

**Review:**

[Summary]
In this paper, the authors proposed a pre-training strategy for Conversational Semantic Parsing (CSP) tasks. The pre-training is run on top of any existing LM (i.e., in this work RoBERTA has been used), and uses three additional loss functions to inject the CSP inductive bias into the LM: Column Contextual Semantics (CCS), Turn Contextual Switch (TCS) and Masked Language Modeling (MLM). Moreover, the authors proposed to use synthetically generated data in the pretraining. The results are presented in four well-know datasets for CSP: SPARC, COSQL, MWOZ, and SQA.

[Pros]
- the proposed pre-training strategy is novel
- the performance of the proposed pre-training strategy is effective

[Cons-Edited]
- the paper claims that "However, existing pre-trained LMs that use language modelling training objectives over free-form text have limited ability to represent natural language references to contextual structural data.", there the authors have not compared the proposed strategy with large pre-trained LMs, especially for Seq2Seq training (e.g., BART, T5, GPT-2) and larger versions of these models. Independently by the number of parameters, which for instance has never been mentioned in the paper, LM trained on text-only can achieve similar or better performance in CSP (Check Author Response Comments), without the need of task-specific pre-trained (i.e. CCS and TCS).
- although few samples, only 500 samples or the dev set only, are used for generating the synthetic data, some of the datasets are used for the pre-training strategy. Moreover, there is a substantial human bias in the construction of the synthetic data, for instance, to create these data probably a human would need to read way more than 500 samples, or even with 500 samples, a human can pretty much guess the distribution of the data, especially for a grammar-based generation as SQL
- the comparison made in the paper are over existing model instead over existing pre-training strategy, or larger models.

[Minor-Cons]
- In Eq. (1) the "SCORE" function is actually a RoBERTA encoder, if I understood correctly, else, this function is not defined anywhere. Why not using  RoBERTA or LM instead?
- In Eq. (1) there is a typo I guess $v_t$ should be $h_t$
- The explanation of the two pre-training loss CCS and TCS is very hard to understand, and Figure 2 doesn't help. I suggest showing more examples.


[Reason to reject] The claim of the paper is not supported for lack of comparisons with different, larger and using different pre-training strategies, LMs. Moreover, the community should be encouraged to create as general as possible pre-trained models, instead of task-specific ones, and especially pre-trained models that use real and unlabeled data.

---

> ### Author Response · Authors · 2020-11-14
> **Author Response**
>
> Thanks for your feedback!
>
> **Comparisons with other existing large language models**
>
> We think our current comparison in the paper is fair and reasonable for the following reasons: (1) Our pre-training approach can be applied to any existing LMs including BART and we demonstrate that it improves the performance on CSP tasks when using BERT/RoBERTa as our base. (2) The reason why we choose BERT/RoBERTa as our base LM is because almost all of the current SOTA models on CSP tasks actually use them. We are not aware of any published results that use BART, T5, or other large LMs for conversational text-to-SQL.
> 1. Regarding your suggestion on comparison with BART, etc., could you please clarify whether you mean comparing base systems (e.g. RAT-SQL) + BERT/SCoRe with (1) BART (cast the task as a seq2seq generation task and directly use BART as the task model), or (2) with base systems (e.g. RAT-SQL) + BART encoder? We agree that both comparisons are interesting to have. But they cannot directly show the effectiveness of our proposed pre-training strategies (without the same base LM and base systems, and involving other unrelated factors). In particular, the first comparison is more like comparing BART vs. Downstream task-specific models (e.g. RAT-SQL).
> 2. ToD-BERT actually uses both MLM and response contrastive loss, not just MLM. For this comparison, we will try to add the experimental results of RAT-SQL + ToD-BERT vs. RAT-SQL + SCoRe on SParC and Trippy + ToD-BERT vs. Trippy + SCoRe on MWoZ before the rebuttal ends.
> 3. We already provide results of SCoRe pre-trained using only MLM (without the CCS and TCS) on all four tasks in Table 6.
>
> **Fairness of pre-training datasets and comparisons**
>
> Thanks for this point. We believe it is unwarranted, for four reasons:
> 1. Most importantly, we only pretrain with MLM on natural data, and with CCS+TCS only on synthetic data (Eq 5). We don’t actually use any labeled data from the same datasets as the evaluation for training. The only exception is using ~500 examples from SParC to manually induce synthetic data grammars for CCS and TCS (Section 2.3).
> 2. Table 6 shows an experiment where only CCS and TCS are used, thus no natural data from downstream datasets are leveraged at all. SCoRe still achieves a significant improvement.
> 3. The SQA dataset is entirely separate as it’s weakly supervised (no SQL labels) yet we pretrain on synthetic text-to-SQL. No SQA data points are seen in any pretraining steps.
> 4. Finally, our comparison with BERT/RoBERTa is fair because it is an important research problem to adapt pretrained language models using in-domain task-related data (e.g., Don’t Stop Pretraining: Adapt Language Models to Domains and Tasks). In Table 6 we did have a comparison with RoBERTa/BERT using MLM-only pretraining, which shows the benefit of SCoRE on four datasets.
>
> **Ablation Study**
>
> We do have an ablation study. In Table 6, we show the effectiveness of three different independent objectives. More discussions are available in the paragraph to answer the research question "What is the effect of each pre-training objective?". In particular, we found SCoRE outperforms RoBERTa/BERT + MLM on additional in-domain data.
>
> **Minor-Cons**
>
> Thanks for catching these! We will submit a revision before the rebuttal deadline by changing "SCoRe" function to "RoBERTa", correcting the typo in Eq 1, and showing more examples of CCS and TCS.

---

> > ### Comment · AnonReviewer5 · 2020-11-14
> > **Response and Clarification**
> >
> > Thanks for your response, let me clarify my review.
> >
> > ***Comparisons with other existing large language models***
> > Let some reason why comparing with different and larger LM is important
> > (1) yes, you can apply this method to any kind of LM, after all, is a pre-training strategy. However, the point here is that if you use large and more powerful LMs (e.g., BART T5 etc.) the advantage of using CCS and TCS may become more and more marginal, and thus a general LM is still preferable over a custom model for this task, which also requires dataset-specific (collected specifically for the task) and synthetic data.
> > (2) The concern I am raising is the effectiveness of the proposed pre-training over different kind of pretraining and larger models, not about SOTA models. Since you are proposing a pre-training strategy, the comparison is among pre-training strategy, and since larger and large LM, especially for Seq2Seq ask, works very well, it is important in my perspective to verify that the give pre-training is really effective.
> >
> > (1 from the list). I mean (1) BART (cast the task as a seq2seq generation task and directly use BART as the task model).
> > R: "But they cannot directly show the effectiveness of our proposed pre-training strategies", I meant running BART as a baseline even without your pre-training strategy, to compare different pre-training strategies.
> > (2 from the list) That's nice, thanks.
> > (3 from the list) I apologies, I did not notice Table 6, thank for point it out. As we can see also here, the adding CCS and TCS lead to marginal improvements, especially considering the amount of annotation needed for CCS and TCS. PS. I  suggest to bold the best in the table, to improve readability.
> >
> > ***Fairness of pre-training datasets and comparisons***
> > (1) As you mentioned, MLM sees samples from the evaluated datasets, this will bias the model over these samples, even if it is just the text. For instance, it has at least seen all the entities and can learn a good representation of those. This would not bias too much the model, but it is important to show results for a completely unseen dataset. After the model is released, it will be fine-tuned to other tasks, what are the expected performance? I mean this would mean simply to finetune the proposed model to another dataset, for example, SGD (Rastogi et.al. 2019).
> >
> > (2) thanks, but this is partially not true. Generating synthetic data as in Campagna et al. (2020) uses the inductive bias from the dataset itself (e.g., MWoZ) to generate the data. Else, the synthetic data would be completely random.
> >
> > (3) and indeed, it doesn't perform as well as other baselines
> >
> > (4) I absolutely agree with your statement, but I don't get the novelty of showing that MLM works in CSP. And again, I am not convinced that pre-training model for seq2seq problems (e.g. BART, T5 or even GPT2-XL) cannot work well in these tasks.
> >
> >
> > In general, two points:
> > - I think pre-training, by definition, should be done over largely available datasets from real data samples. Otherwise, the resulting model will be helpful only in particular domains and problems.
> > - The baselines, although SOTA, are not relevant to the claim, and to it is not convincing a so complex pre-training (CLS and TCS) vs over simple MLM or even MLE as in GPT. BERT and RoBERTA are good in classification tasks, CSP is more a generation task in my understanding.
> >
> >
> > I hope my comments do not sound rude or aggressive, they are not meant to be. I like the CSP approach to ToDs, and I happy to see such progress in this field.
> >
> > Thanks again for your response, I hope my comments can engage in more discussions.
> >
> > Looking forward to hearing from you.

---

> > > ### Author Response · Authors · 2020-11-15
> > > **Author Response**
> > >
> > > Thanks a lot for your clarification and comments!
> > >
> > > **Do existing LMs (BART, T5, GPT-2) outperform custom models + BERT on CSP tasks?**
> > >
> > > ***Mostly No, based on our experiments and other published results.*** We cannot find any published results on SParC/CoSQL using BART/T5. However, we have found some evidence on Spider. Spider is the single-turn version of SParC and CoSQL. Among >70 submissions, only 3 or 4 of them use BART/T5 (they are not SOTA) and most use BERT/RoBERTa. If BART or T5 can simply be applied to CSP tasks, it is very likely that they already SOTA and would be used more often. Below, we compare T5/BART with RAT-SQL+BERT-Large, which is the same SOTA base system for SParC and CoSQL in our paper.
> > >
> > > (1) T5. From a recent Google paper [Compositional Generalization and Natural Language Variation: Can a Semantic Parsing Approach Handle Both?](https://arxiv.org/abs/2010.12725), in Table 4 they applied T5 as seq2seq to Spider:
> > >
> > > | Model(parameters) | Spider |
> > > |---|---|
> > > |T5-Base (220M)| 57.1|
> > > | T5-3B (3000M)| 70.0|
> > > | RAT-SQL + BERT-Large (~500M) | 69.7|
> > >
> > > RAT-SQL+BERT-Large outperforms T5-Base by 12.6. ***T5-3B improves only 0.3, but it is 6 times larger*** and cannot be used on usual GPUs (Google can use TPU), so it is prohibitively expensive for us. SCoRe is significantly smaller than T5-3B and hence easier, cheaper, and faster to finetune and deploy.
> > >
> > > (2) BART. During the beginning of our project, we have performed experiments to compare BERT, BART encoder, and BART encoder+decoder as seq2seq. ***We found BART cannot outperform custom models + BERT***:
> > >
> > > | Model|Spider|
> > > |-|-|
> > > |RAT-SQL + BERT|69.7|
> > > |RAT-SQL + BART encoder|67.8|
> > > |BART encoder + decoder (406M, as a seq2seq task)|62.4|
> > >
> > > Moreover, from Table 2 and 3 in [SMBOP: Semi-autoregressive Bottom-up Semantic Parsing](https://arxiv.org/abs/2010.12412),  BART didn't outperform BERT:
> > >
> > > |Model|Spider|
> > > |-|-|
> > > |RAT-SQL + BART-Large (No DB used)|66.0|
> > > |RAT-SQL + BERT-Large (DB used)|69.7|
> > > |RYANSQL + BERT (No DB used)|60.6|
> > > |SMBOP + BART-Large   (No DB used)|60.5|
> > >
> > > (3) GPT-2. From Table 5 in ToD-BERT and Table 1 in SimpleToD, ***GPT-2 does not outperform BERT on MWoZ***:
> > >
> > > |Model| MWoZ|
> > > |-|-|
> > > |GPT-2 (ToD-BERT)|46.2|
> > > |BERT-Base (ToD-BERT)|45.6|
> > > |GPT-2 (SimpleToD)|56.5|
> > > |TripPy BERT|58.4|
> > >
> > > **Are our improvements marginal?**
> > >
> > > ***The improvements of our CCS+TCS without MLM are actually very significant compared to MLM only.*** The best results are achieved using CCS+TCS without MLM on SParC, CoSQL, and SQA. We believe the improvements of ~2-6% on these competitive tasks are not marginal. From Table 6,
> > > 1. CCS+TCS largely outperforms MLM only (+5.5% on SParC, +1.7% on CoSQL, and +3.4% on SQA)
> > > 2. Adding MLM to CCS+TCS hurts the performance (-3.9% on SParC, -0.3% on CoSQL, and -4.4% on SQA)
> > > 3. Using only MLM (SCoRe MLM only vs. RoBERTa) helps a bit (<~1%) compared to RoBERTa
> > >
> > >
> > > **Fairness of pre-training datasets and comparisons**
> > >
> > > As mentioned above, the best results are achieved by pre-training SCoRE on ***only synthesized data (using CCS+TCS) without any natural questions (using MLM)*** on SParC, CoSQL, and SQA. We do NOT highlight MLM as our contribution. Actually, we show that adding MLM to TCS+CCS hurts. To clarify our usage of datasets,
> > > 1. CCS+TCS doesn't use any label data except only ~500 SParC examples to induce synthetic data grammars (Section 2.3).
> > > 2. NO CoSQL or SQA data are seen in our pre-training, and SCoRe with CCS+TCS still outperforms RoBERTa on CoSQL (+2.7%) and SQA (+4.9%).
> > > 3. For WMoZ, Campagna et al. (2020) use only the *dev* examples to induce the grammar.
> > >
> > >
> > > **Should pre-training be done over only largely available datasets from real data samples?**
> > >
> > > ***The difficulty is CSP data annotation is VERY expensive***. Semantic parsing tasks are different from text-to-text generation because it has to also encode other inputs (such as structural database schema and table content) and then decode formal programs (such as SQL). Therefore, using larger models (BART, T5, GPT-2) as a seq2seq model for semantic parsing is not a trivial task. We agree that ideally pre-training for CSP should be done over large real CSP datasets. However, CSP annotation requires experts to annotate formal programs such as SQL. Even by combining existing CSP datasets, the size is still limited.
> > >
> > > ***As our contribution, we prove how to perform effective pre-training on synthesized data for many CSP tasks*** with conversational and compositional questions. In fact, while data synthesis has been widely applied ([Berant and Liang, 2014], [Wang et al., 2015], [Jia and Liang, 2016], [Andreas, 2020]), how to use them for pre-training is still unclear. We demonstrate the effectiveness of our pre-training objectives (TCS+CCS) over multiple representative CSP tasks including fully- and weakly- supervised (dialog state tracking, context-dependent semantic parsing, SQL-grounded state tracking, sequential question answering).

---

> > > > ### Comment · AnonReviewer5 · 2020-11-16
> > > > **Re: Comments**
> > > >
> > > > ***Do existing LMs (BART, T5, GPT-2) outperform custom models + BERT on CSP tasks?***
> > > >
> > > > 1) Thanks for reporting these numbers, this actually confirms the point I was making at the beginning in my review. Larger LMs, e.g. T5, which are pre-trained with ***text only***, not even related to CSP or dialogue, can perform well in this task. And "cannot be used on usual GPUs" is actually not true (https://huggingface.co/t5-3b), for the pre-trained of SCoRe you have been using 8 V100s, with a little engineering T5 can be run in probably two V100, with probably very few finetuning steps needed. Anyhow, the main point is that: independently by the number of parameters, which for instance has never been mentioned in the paper, LM on text-only can achieve similar or better performance in CSP. Especially, given that one of the main claims/motivation of the paper is:
> > > > "However, existing pre-trained LMs that use language modelling training objectives over free-form text have limited ability to represent natural language references to contextual structural data"
> > > > Why shall we prefer a custom architecture or very complex custom pre-training strategy over LM pre-training? notice number of the parameter can be reduced with compression techniques for example, and again the paper never mentions model size.
> > > >
> > > > 2) I am a bit confused by the table, BART with (No DB used) seems to perform very well. I am wondering if is there a result with BART + DB used.
> > > >
> > > > 3) From my understanding, ToD-BERT is a classification approach for DST, thus, of course, GPT doesn't perform well. And in Simple ToD, the GPT is trained to generate all the Dialogue-State, so it needs also to learn the slot names. Anyhow, also here, the size of the model matters if we talk about SOTA, and in GPT, the larger the better.
> > > >
> > > > ***Are our improvements marginal?***
> > > > Thanks for clarifying, the increase in percentage is subjective, and since we are out for subjective comments,  I believe that for a much simpler pre-training such as MLM, or a general pre-trained model, the reported improvement is marginal. But as I mentioned this is subjective, depending on what is the experimental setting.
> > > >
> > > > a) fairness of pre-training datasets and comparisons: synthetic data generation is tricky because, yes you use only 500 samples or the dev set only, but there is a huge human bias in the construction of these datasets, which probably read way more than 500 samples, or even if you read 500, can pretty much guess the distribution of the data, especially for a grammar-based generation as SQL.
> > > >
> > > > ***Should pre-training be done over only largely available datasets from real data samples?***
> > > > Thanks for this comment, I mostly agree.
> > > >
> > > > ***Contribution***
> > > > Thanks for your summary, I would suggest to include this more clearly in the paper.
> > > >
> > > >
> > > > Thanks again for your comments. I believe we mostly clarify the doubts.
> > > > I will update my score and my review accordingly, and let the AC and other reviewers decide. I suggest making a new comment summarizing this discussion, for facilitating the AC job and having your paper fairly judge.

---

> > > > > ### Author Response · Authors · 2020-11-20
> > > > > **Author Response**
> > > > >
> > > > > Thanks again for your comments.
> > > > >
> > > > > **1. Result interpretation**
> > > > > We are afraid we do not agree with how the reviewer interpreted the results we shared in the comments. The comments mentioned two methods:
> > > > > - (a) A very large LM finetuned on text-to-SQL (T5-3B, ~ 3000M parameters)
> > > > > - (b) A custom model (RAT-SQL) + BERT-large finetuned on text-to-sql task (~ 500M parameters).
> > > > >
> > > > > Both (a) and (b) have similar performance (within 0.3 points) as reported in Shaw et al.  We use (b) as our baseline and show that replacing BERT/RoBERTa with SCoRe can result in 5.8% improvement on SParC, 2.7% improvement on CoSQL and 4.9% on SQA.
> > > > > While we do not directly compare our approach to T5. We show that our approach >> (b) and (a) ~= (b). As such, *we think the statement in the review “LM trained on text-only can achieve similar or better performance in CSP” is only justified when comparing T5-3B to BERT* ***but not justified when comparing T5-3B to the proposed model in this paper SCoRe***.
> > > > >
> > > > > Whether going to even larger models (e.g.10s or 100s  billion parameters) may erase any gain that a method like SCoRe may achieve is an open research question that we do not have an answer for now. This is also a question that is relevant to pretty much every other NLP task out there.
> > > > >
> > > > > [Another recent EMNLP’20 paper](https://arxiv.org/pdf/2002.08910.pdf), from a subset of the authors of T5, provides more evidence that larger LM is not sufficient for the task of open domain question answering. Their findings are:
> > > > > - An 11B parameter model (T5) underperforms SOTA models (SOTA is a custom model +BERT) on three datasets (Natural Questions: 41.5 vs. 32.8, WebQuestions: 42.4 vs. 42.8, TriviaQA: 57.9 vs. 42.9)
> > > > > - The authors ***add a second-stage of pre-training using a QA specific objective*** (salient span masking) to T5-11B and the results improve but still not SOTA  (Natural Questions: 35.2, WebQuestions: 42.8, TriviaQA: 51.9). This further supports our hypothesis that adding a “second stage of pre-training to induce inductive bias beyond general LM objectives is useful for many tasks”.
> > > > >
> > > > > 2.Overall, ***we agree that exploring if larger LMs pre-trained on text only can perform well in CSP or dialog tasks is a very interesting question***. This is part of an even bigger question on the role of custom models that aim to build innate priors into the models for solving NLP tasks vs. fully relying on one large pre-trained LM.  **This is a heavily debated topic and the NLP community is far from reaching any conclusive outcome. We believe there is value in pursuing both directions and being on one side of the debate shouldn’t be grounds for dismissing work on the other side**.
> > > > >
> > > > > We agree with the reviewer that we have shown the benefits of our approach over LMs in the size of 100s M of parameters (BERT/RoBERTa) but did not show their benefit for even larger models with billions of parameters (e.g. T5).  We chose to use BERT/RoBERTa because: (1) it is used by all SOTA methods over the 4 tasks we experimented with and (2) evidence from recent work (e.g. Shaw et al.) suggests that much larger models do not outperform custom models + BERT. While we believe the approach will be valuable even for extremely larger models (e.g. T5), we acknowledge that this is an empirical question for future work.
> > > > >
> > > > > 3.We would also like to point out that we believe that **accuracy is not the only measure, model size matters and is of interest to the research community**. Even if a custom task-specific model (e.g., RAT-SQL+BERT) achieves the same accuracy as much larger LMs (e.g., T5-3B, or T5-11B), the smaller models present significant economical and environmental benefits and will ensure the advances are available to a much wider audience. Using the smaller model will be the clear (and often only) choice available to many companies and universities that cannot afford to use much larger models (e.g. T5) for deployment or research. For practical reasons, and even if a company can afford the more expensive model, they would also favor a smaller model that is as performant to reduce their deployment costs or to serve the model in a resource-constrained setting (e.g. on a mobile device).  This is a very actively discussed topic, in NLP and other applications of ML.
> > > > >
> > > > > Regarding the comment on compression, while this is a very promising research direction, compression usually comes at a cost of performance loss. Additionally, compression is not only applied to extremely large models (e.g. T5) but can also be applied to large models like Bert/SCoRe.
> > > > >
> > > > > 4.**Fairness of pre-training datasets and comparisons**: That is why we also tested the effectiveness of SCoRe on two totally unseen datasets CoSQL and SQA (weakly-supervised SCP without SQL labels). The improvements of SCoRe on CoSQL and SQA show that the proposed pre-training method is general even when pre-training data is only synthesized by a grammar induced by SParC examples.

---

### Author Response · Authors · 2020-11-25
**Summary of discussions with R5**

We summarize the discussion that we had with R5 here for the benefit of the other reviewers and the AC. We would like to thank R5 for the detailed discussion and for suggesting that we share a summary of it with everyone.

**Using extremely large pretrained LMs (e.g. T5) as baselines**

Our own experiments with BART (Appendix E) and results of using T5 for semantic parsing (Shaw et al.) and question answering (Roberts et al.) show than using large pretrained models like underperforms or yields comparable performance to SOTA models (custom models + BEERT/RoBERTa,  e.g. RATSQL+BERT). We opt for using the latter as baselines since it is the SOTA published results and are smaller in size, easier to train, deploy, etc.

**Task-specific pre-training v.s. larger LM models**
1. We agree that exploring whether larger LMs pre-trained on text only can perform well in CSP or dialog tasks is a very interesting question. This is part of an even bigger question on the role of custom models that aim to build innate priors into the models for solving NLP tasks vs. fully relying on one large pre-trained LM.
2. This is a heavily debated topic and the NLP community is far from reaching any conclusive outcome. We believe there is value in pursuing both directions and being on one side of the debate shouldn’t be grounds for dismissing work on the other side.
3. We also argue that accuracy is not the only measure, model size matters and smaller model sizes are of interest to the research community due to economical, environmental and wide accessibility concerns.
4. Finally, we note that our model is not specific to one task or one dataset. Our experiments show it improves the performance of different base models on 4 different CSP tasks (fully-sup., weakly-sup, and dialog state tracking, QA, semantic parsing).

**Fairness of using 500 examples from SPARC in the synthetic data generation**
1. We ran additional experiments as suggested by R3 (see Appendix E). Our results show than adding these 500 examples in the training process yields no or very small gains.
2. Note that in our original experiments, we tested the effectiveness of SCoRe on two totally unseen datasets CoSQL and SQA (weakly-supervised SCP without SQL labels) to show that the benefits of the proposed pre-training method is general.

**Improvements of SCoRe over the four CSP tasks are substantial**

We agree that the increase in percentage is subjective, but we can still find some objective evidence by looking at prior improvements on these competitive tasks.
1. As you can see on [MWoZ public leaderboard](https://github.com/budzianowski/multiwoz#belief-tracking), a significant improvement in SOTA as defined by most recently published works is about 1.5%. SCoRe improves 2.1% over BERT and 1.8% over the very recent prior SOTA.
2. On [SParC leaderboard](https://yale-lily.github.io/sparc), a significant improvement is about 2%, and SCoRe outperforms prior SOTA by 3.7% in QM and 4.8% in IM. Similarly, on [CoSQL leaderboard](https://yale-lily.github.io/cosql), a significant improvement is also 2%, and SCoRe outperforms prior SOTA by 4.8% in QM and 4.2% in IM.

**Other concerns raised by R5  (that has been addressed has since been deleted from the edited review after the discussion)**
1. ToD-BERT: we clarified that ToD Bert does in fact use a response contrastive loss, not just MLM. We added a comparison against ToD Bert in Appendix E.
2. Ablation study: we pointed to several ablation studies and added more as per the discussion (Section 4 and Appendix E).
3. Pre-training dataset: we clarified that we do not use the labeled data from any of the datasets for pre-training.
4. We addressed other minor issues (see list of paper edits for details).

---

### Author Response · Authors · 2020-11-25
**Summary Response to All Reviewers**

We thank all reviewers for their thoughtful feedback. We are glad that the reviewers appreciate the novelty and the effectiveness of our proposed approach (R5), find our experiments to be comprehensive and convincing by achieving SOTA on 3 out of 4 different tasks (R1, R2, R3, R4), ablation studies and analysis to be informative and well done (R2, R4), and think our paper is clearly written and easy to follow (R1, R2, R3, R4).

We have updated our paper with the following changes to reflect the reviewer comments:

- update ablation study of different objectives - pre-training with CCS+TCS (only synthesized data used) objectives achieves the best performance on three tasks (Table 6), which is very significant compared to MLM. (Section 4 - What is the effect of each pre-training objective?)

Reviewer3 and Reviewer4
- clarify the usage of datasets in pre-training with different objectives (MLM vs. CCS+TCS vs. CCS+TCS+MLM). Basically, (CCS+TCS) is pre-trained on only synthesized data (Section 3.3)

Reviewer4 and Reviewer5
- BART Encoder+Decoder on Spider (Appendix E) - using BART Encoder+Decoder on Spider underperforms SOTA model (RATSQL+BERT)

Reviewer 5
- change "SCoRe" function to "RoBERTa" (Section 2.1)
- correct the typo in Eq 1 (Section 2.1)
- ToD-BERT (Appendix E) - RAT-SQL + SCoRe outperforms RAT-SQL + ToD-BERT by 7.6%

Reviewer 2
- TCS experiments (Appendix E) - TCS only improves 2.4% so far on SParC
- comparison with Herzig et al. (2020) on SQA (Appendix C.2)
- add analysis of few-shot learning with 10% training data on SQA as a separate paragraph (Section 4)

Reviewer 3
- data concatenation or multi-task training results (Appendix E) - incorporating the additional SParC examples does not significantly improve the performance on CoSQL and SQA compared with SCoRe.
- update equations 2 and 3 by including the formal representation q (section 2.2)

Reviewer 1
- improve discussion on motivation in the introduction (section 1)
- comparisons with related work on other contextualization methods (section 5)
- clarification on TCS and spacing (section 2.2)
- move Table 1 to separate it from Figure 1

Reviewer 4
- add more data synthesis details (section 2.3)
- add citations in Table 3
- correct statistics about MultiWOZ 2.1 (Section 3.1 and caption in Table 1)
- add more description of SQA (Section 3.1)
- add more details on pre-training steps (section 2.2-Pre-Training Setup and Steps)

---

### Decision · Program_Chairs · 2021-01-07
**Final Decision**

**Decision:**

Accept (Poster)

**Comment:**

This paper proposes to pre-train contextual semantic parsing models on synthesized data (using a small amount of additional supervised training data and grammar-based generalizations therefrom) with two new training objectives: Column Contextual Semantics (CCS), mapping text to database columns, and Turn Contextual Switch (TCS), to deal with the update semantics between turns.

I thank the reviewers for their detailed engagement with this paper, and thanks the authors for their responsiveness in doing extra experiments and rewriting that made this paper better and the decision clearer.

Pros

The authors did such a great job of summarizing the pros, that I think I can just copy their summary: "We are glad that the reviewers appreciate the novelty and the effectiveness of our proposed approach (R5), find our experiments to be comprehensive and convincing by achieving SOTA on 3 out of 4 different tasks (R1, R2, R3, R4), ablation studies and analysis to be informative and well done (R2, R4), and think our paper is clearly written and easy to follow (R1, R2, R3, R4)."

Cons

- A somewhat specific and ad hoc data synthesis solution
- Stronger pre-trained contextual language models might beat assumed baselines or methods shown here (R4, R5)
- The story is weak and should be better motivated through discussion of contextualization of interpretation

In general the reviewers recommend accepting the paper, and I agree. However, it is perhaps not of the novelty, clarity, or impact size to qualify for more than a Poster. R5 has a good point about how strong pre-trained LMs are a general tool and should be preferred to the extent they work in 2020, but I think they are too opinionated to suggest this is a reason for rejection. Along with the other reviewers and the authors, I think it is most reasonable to accept work showing good progress using "medium-sized" pre-trained LMs -- really we thought BERT was big a couple of years ago! -- and this work has comprehensive experiments with good results. I would encourage the authors:

- To say more about the alternative strategy of instead using a bigger pre-trained LM, as has come out in the discussion on OpenReview, and the pros and cons of this approach (though maybe the results with BART are the only fairly comparable data point)
- To strengthen the presentation by orienting the paper more around the importance of contextualization in interpreting dialog turns in conversational semantic parsing (as opposed to the "one turn" nature of the original famous semantic parsing datasets).

p.s. One typo I noticed in the revised paper while reading: fours --> four